# Dietary Cadmium Intake and Sources in the US

**DOI:** 10.3390/nu11010002

**Published:** 2018-12-20

**Authors:** Kijoon Kim, Melissa M. Melough, Terrence M. Vance, Hwayoung Noh, Sung I. Koo, Ock K. Chun

**Affiliations:** 1Department of Food and Nutrition, Sookmyung Women’s University, Seoul 04310, Korea; drkijoon@gmail.com; 2Department of Nutritional Sciences, University of Connecticut, Storrs, CT 06269, USA; melissa.melough@uconn.edu (M.M.M.); sung.koo@uconn.edu (S.I.K.); 3Department of Nutrition and Dietetics, SUNY College at Plattsburgh, Plattsburgh, NY 12901, USA; tvanc001@plattsburgh.edu; 4International Agency of Research on Cancer, World Health Organization, 69372 Lyon, France; nohh@fellows.iarc.fr

**Keywords:** cadmium, diet, NHANES

## Abstract

Cadmium (Cd) is a toxic heavy metal that can contribute to numerous diseases as well as increased mortality. Diet is the primary source of Cd exposure for most individuals, yet little is known about the foods and food groups that contribute most substantially to dietary Cd intake in the US. Therefore, the objective of this study was to estimate dietary Cd intake and identify major food sources of Cd in the US population and among subgroups of the population. Individuals aged 2 years and older from the National Health and Nutrition Examination Survey (NHANES) 2007–2012 were included in this study (*n* = 12,523). Cd intakes were estimated from two days of 24-h dietary recalls by matching intake data with the Cd database of the Food and Drug Administration (FDA)’s Total Diet Study 2006 through 2013. The average dietary Cd consumption in the population was 4.63 μg/day, or 0.54 μg/kg body weight/week, which is 22% of the tolerable weekly intake (TWI) of 2.5 μg/kg body weight/week. Greater daily Cd intakes were observed in older adults, males, those with higher income, higher education, or higher body mass index. The highest Cd intakes on a body weight basis were observed in children 10 years and younger (38% of TWI), underweight individuals (38% of TWI), and alcohol non-consumers (24% of TWI). The food groups that contributed most to Cd intake were cereals and bread (34%), leafy vegetables (20%), potatoes (11%), legumes and nuts (7%), and stem/root vegetables (6%). The foods that contributed most to total Cd intake were lettuce (14%), spaghetti (8%), bread (7%), and potatoes (6%). Lettuce was the major Cd source for Caucasians and Blacks, whereas tortillas were the top source for Hispanics, and rice was the top contributor among other ethnic subgroups including Asians. This study provides important information on the dietary Cd exposure of Americans, and identifies the groups with the greatest dietary Cd exposure as well as the major sources of dietary Cd among sociodemographic subgroups.

## 1. Introduction

Cadmium (Cd) is a toxic heavy transition metal released as a result of industrial and agricultural activities into soil and water where it can be absorbed by and accumulated in plants and aquatic organisms destined for the food supply [1]. Our research group has recently reported that smoking is strongly associated with Cd exposure levels measured in blood and urine [2,3], and that the interactions of Cd with essential minerals such as zinc differs by smoking status [4]. Although tobacco smoke is an important source of Cd exposure among smokers [5], and certain occupations have high risk of Cd exposure, diet is the main source of Cd exposure for most people [1]. Due to the chronic nature of dietary Cd exposure, combined with the long half-life of Cd in the human body, Cd can accumulate in multiple tissue types, contributing to the development of cancer [6,7,8], kidney dysfunction [9,10], cardiovascular disease [11], reproductive dysfunction [12,13], diabetes [14], osteoporosis [15], and increased mortality [16].

Given that diet is one of the most relevant sources of Cd exposure, determination of the major dietary Cd contributors and identification of population groups with high dietary Cd intakes are critical public health priorities. Estimates of dietary Cd intake and major dietary Cd sources are available in the published literature for a few selected countries [17,18,19,20]. These reports demonstrate that Cd intake levels, as well as the primary sources of dietary Cd, can vary between and within countries, depending on dietary patterns and the levels and areas of Cd contamination in different food environments. No systematic investigation has been undertaken using detailed dietary data to estimate dietary Cd exposure in the US population and its sociodemographic subgroups, or to identify major Cd sources in the American diet. While one study estimated dietary Cd exposure among a cohort of postmenopausal US women [21], no study has reported the dietary Cd exposure among other subgroups of Americans or thoroughly examined the major dietary sources in the American diet. Therefore, the objective of this study was to estimate dietary Cd intake of the US population and its sociodemographic subgroups, and to document the major sources of dietary Cd among the US population using the National Health and Nutrition Examination Survey (NHANES) 2007–2012.

## 2. Materials and Methods

### 2.1. Study Population

This cross-sectional study was conducted using data from individuals aged 2 years and older from NHANES 2007–2012. We excluded individuals who had not completed two days of dietary recalls (*n* = 4952), those who reported consuming breast-milk (*n* = 11), those whose recalls were coded as unreliable or incomplete (*n* = 182), those whose dietary recalls represented consumption that was “much more than usual” or “much less than usual” (*n* = 9393), and those who reported being on any kind of special diet (*n* = 2423), so as to obtain a sample of individuals whose dietary data were representative of usual intake. After these exclusions, the analytic cohort consisted of 12,523 individuals.

### 2.2. Estimation of Dietary Cd and Sources

Dietary data were collected from two 24-h dietary recalls from NHANES 2007–2012. The US Food and Drug Administration (FDA) Total Diet Study (TDS) is an ongoing study that monitors and reports the concentrations of chemical contaminants in the US food supply [22]. We used data from the TDS in the 2006 through 2013 market baskets, which include the Cd concentrations of 260 individual foods [23]. Daily Cd intakes and weekly Cd intake per kg of body weight were estimated from two days of dietary recalls for each participant by matching dietary consumption data with the Cd data in the TDS. We used the USDA Food and Nutrient Database for Dietary Studies (FNDDS) version 4.1 (2007–2008) [24], the FNDDS version 5.0 (2009–2010) [25], and the FNDDS 2011–2012 [26] to assign Cd concentrations to the ingredients of complex food items reported in NHANES 2007–2008, NHANES 2009–2010, and NHANES 2011–2012, respectively. The Cd concentrations of these food ingredients were summed to estimate Cd contents of complex foods. The Cd contents of foods were also adjusted for moisture and fat changes that occur during cooking. Total Cd intakes were estimated by summing contributions from all foods reported. Dietary Cd contribution from dietary supplements was not considered in this analysis due to the lack of information on the Cd contents of dietary supplements in the TDS. To determine the top food sources contributing to Cd intake, 154 individual food items were extracted based on foods for which Cd contents were available in the TDS. Fifteen food groups, including non-allocated foods, were mutually exclusively created for the analysis of major food group contributors to Cd intake. Non-allocated foods were those that did not fit into traditional food groupings such as sauces and baby foods. The top contributing individual food items and the percent contribution to total Cd intake of each food were determined for the US population and among population subgroups.

### 2.3. Statistical Analysis

Statistical analyses were conducted using SAS software, version 9.4 (SAS Institute Inc., Cary, NC, USA), using SAS survey procedures and the appropriate weight, strata, and cluster variables to account for the complex survey design of NHANES. Mean and 95% confidence intervals (CIs) of daily Cd intake and weekly Cd intake per kg body weight were calculated across groups by sociodemographic and lifestyle characteristics. *p*-values for differences in daily Cd intake and weekly Cd intake per kg body weight between subgroups were obtained by *t*-test and ANOVA. Participants were grouped by poverty income ratio (PIR) as follows: PIR < 1.0, 1.0 ≤ PIR < 1.3, 1.3 ≤ PIR < 1.85, 1.85 ≤ PIR < 3.5, and PIR ≥ 3.5. Based on the number of drinks of any type of alcoholic beverage per day, alcohol consumption was defined as no consumption (0 drinks), moderate consumption (no more than 2 drinks/day for men and no more than 1 drink/day for women), and heavy (more than 2 drinks/day for men and more than 1 drink/day for women) [27]. Current smokers were defined as those who smoked at least 100 cigarettes in their lifetime and smoke some days or every day. Non-smokers were defined as those who smoked fewer than 100 cigarettes in their lifetime and serum cotinine level ≤0.05 ng/mL or those who quit smoking over 3 years ago with serum cotinine level ≤0.05 ng/mL. Passive smokers were defined as those who smoked fewer than 100 cigarettes in their lifetime and serum cotinine level >0.05 ng/mL, or those who quit smoking over 3 years ago with serum cotinine level >0.05 ng/mL, or those who inhale the smoke from others’ cigarettes at work or at home and serum cotinine level >0.05 ng/mL. All *p*-values reported are two sided (α = 0.05).

## 3. Results

Average daily Cd intake among the US population was 4.63 μg/day, and weekly Cd intake per kg body weight was 0.54 µg/kg body weight/week (Table 1), which is 22% of the tolerable weekly intake (TWI) of 2.5 μg/kg body weight/week [28]. Greater daily Cd intakes were observed in males, and those with higher incomes and education levels. Lower dietary Cd intakes were observed in current smokers, those who were underweight, non-consumers of alcohol, and those with lower education levels. Higher Cd intake on a body weight basis was noted among males, the youngest age group (2–10 years old), and those who were underweight.

The top food groups contributing to Cd intake in the US population were cereals and bread (34%), leafy vegetables (20%), potatoes (11%), legumes and nuts (7%), stem/root vegetables (6%), and fruits (5%) (Figure 1).

Across all age groups examined, spaghetti, and bread ranked among the top three greatest individual foods contributing to total dietary Cd (Table 2). Lettuce was the top contributor for adolescents (11–19 years old) and adults (20+ years), and was the fourth greatest contributor to total Cd intake among children (2–10 years). Potatoes and potato chips were also among the top Cd contributors across all age groups. Among children aged 10 years and younger, peanuts, cookies, strawberries, and milk were high contributors to total Cd intake in comparison to their contribution among older Americans. Sunflower seeds were relatively greater contributors among adolescents than younger children or adults. Spinach, tomatoes, and beer were greater contributors to total Cd intake among adults in comparison to among children or adolescents. 

Lettuce was the top source of Cd among both Caucasians and blacks, whereas tortillas were the top source for Hispanics, and rice was the top source for other ethnic subgroups including Asians (Figure 2). Bread and spaghetti were rather important Cd contributors across all racial/ethnic subgroups, whereas potato chips were a particularly high contributor among Blacks and Whites, and noodles were a relatively important source among those who identified as Asian or other races.

## 4. Discussion

Our findings indicated that among the US population ages 2 years and older in 2007 through 2012, the latest period for which Cd data were available in the TDS, the average daily Cd intake was 4.63 μg/day. This estimate is lower than some previous estimates among segments of the US population. Using food frequency questionnaire (FFQ) data from postmenopausal women in the Women’s Health Initiative (WHI), along with Cd data in the TDS 1991–1996 market baskets, Quraishi et al. estimated an average dietary Cd intake of 10.4 μg/day among these women [21]. Similarly, using Cd data from the TDS 1998–2008 market baskets linked to FFQ data from postmenopausal women in the VITamins and Lifestyle (VITAL) cohort, Adams et al. estimated dietary Cd to be 10.9 μg/day [29]. Our study differs from these previous reports in the population examined, as well as the method of dietary data collection, which may contribute to the differences in estimated Cd intake. Another potentially important reason for the difference in these estimates may be that the WHI and VITAL studies examined earlier time periods than our study. Among US adults, urinary Cd concentration, which is frequently used as a biomarker of long-term Cd exposure [30], decreased by 34% from 1988–1994 to 2003–2008 [31]. This decline is likely related in part to reduced rates of smoking, yet a large proportion of the reduction is likely to be related to changes in Cd concentrations in the food supply [31]. Reductions in Cd concentrations in the food supply may have resulted from new legislation in certain US states limiting the Cd content of phosphorus fertilizers [32], yet the effects of such laws on human dietary exposure to Cd has not yet been systematically examined.

Current assessments of dietary Cd exposure in the US are generally lacking, and older estimates indicated relatively high exposure compared to more recent estimates. Using dietary data from NHANES III, which was conducted in 1988–1994, one group estimated the geometric mean of daily Cd intake in the US population was 18.9 μg [33]. Another study conducted in the late 1960s indicated that US dietary Cd exposure was 26 μg/day/person [34], and a 1978 report indicated Cd exposure among US adults was 26 μg/day/person [35]. These older studies were based on less comprehensive data on the Cd contents of various foods in the US, and may also differ from more current estimates due to differences in study methods as well as changes in the food supply and/or dietary patterns. 

In this study, we also documented major sources of Cd intake. We found that cereals and breads, followed by leafy vegetables and potatoes, were the top food groups contributing to dietary Cd intake among Americans aged 2 years and older. Similarly, in their examination of postmenopausal women in the VITAL study, Adams et al. found that grains and vegetables including potatoes collectively contributed 66% of total estimated dietary Cd [29]. Among postmenopausal women in the WHI, vegetables accounted for 42% of total dietary Cd, and grains accounted for 29% [36]. These plant foods are vulnerable to Cd contamination from fertilizer use and from other agricultural and industrial activities [1,32], and it is therefore unsurprising that they are consistently found to be major contributors to total dietary Cd.

Existing literature demonstrates the differences in dietary Cd sources across nations with differing populations, dietary patterns, and food environments. In studies conducted in Chile [19], Hong Kong [37], Thailand [38], and Spain [39,40], fish and shellfish were identified as major Cd sources in addition to grains and vegetables. In comparison, our results showed that fish and shellfish were relatively minor sources of Cd intake in the US population. Studies conducted in Belgium [17] and Germany [18] similarly indicated a low contribution of fish and shellfish to total Cd intake.

We also found that the major food items contributing to dietary Cd intake differ by sociodemographic subgroups within the US. For example, peanuts, cookies, strawberries, and milk were relatively more important Cd contributors among young children than among older age groups. Lettuce, spinach, and beer were relatively more important contributors to Cd intake among adults than among children or adolescents. Lettuce was among the top contributors to Cd among all ethnic groups examined, and was the greatest source for whites and blacks. However, tortillas outranked lettuce as a top source among Hispanic Americans, and rice outranked lettuce as the top source among other ethnicities including Asians. Although the concentration of Cd is higher in lettuce (0.066 mg/kg for raw leaf lettuce and 0.051 mg/kg for iceberg lettuce), than in tortillas (0.019 mg/kg) or rice (0.006 mg/kg) [23], tortillas and rice are high Cd contributors among Hispanics and Asians due to the high frequency of consumption, as has been reported previously [41]. Several studies have shown that those with higher education levels and higher incomes consumed more fruits and vegetables [42,43,44]. Because these items are major contributors to dietary Cd intake, it is unsurprising that our results indicated higher daily Cd intake among these groups. 

A major strength of this study is the use of a large, nationally representative sample of the US population, allowing for thorough analysis of Cd intake levels and major food sources among subgroups by age, ethnicity, gender, income, and lifestyle factors. This study also has limitations. First, estimation of dietary Cd intake relied on two days of 24-h dietary recalls, which may not capture all dietary Cd sources, especially from episodically consumed foods. However, the Automated Multiple Pass Method used for dietary data collection in NHANES has been well validated [45,46]. Furthermore, our data are in agreement with previous studies that used FFQs to assess dietary intake [21,29], which reflect longer term intake, suggesting our analysis was suitable for identifying the major sources of Cd in the American diet. Secondly, Cd intake may be underestimated in our study because of the limited number of foods for which Cd data are available in the TDS. However, the TDS likely captures the most important Cd contributors in the US diet. The TDS updates its food lists based on national food consumption surveys and is representative of the major components of the average diet of the US population [22]. Importantly, because Cd intake in the US may change over time due to changes in consumption patterns and/or levels of contamination in the food supply, our findings are specific to US dietary Cd exposure from 2007 to 2012. Finally, it is important to note that dietary Cd bioavailability can be affected by the food source and co-ingested foods [47], and may also be influenced by individual characteristics including nutritional status [48]. Findings from this study can be used in future work examining Cd burden in the US population, but should be carefully interpreted because of the potential differences in gastrointestinal Cd absorption based on dietary sources and host factors.

## 5. Conclusions

In conclusion, this study documented the daily Cd intake level and major food sources of the US population by age and ethnic subgroups. These data may provide a foundation for future studies investigating Cd burden and the relationships between Cd exposure and health status.

## Figures and Tables

**Figure 1 nutrients-11-00002-f001:**
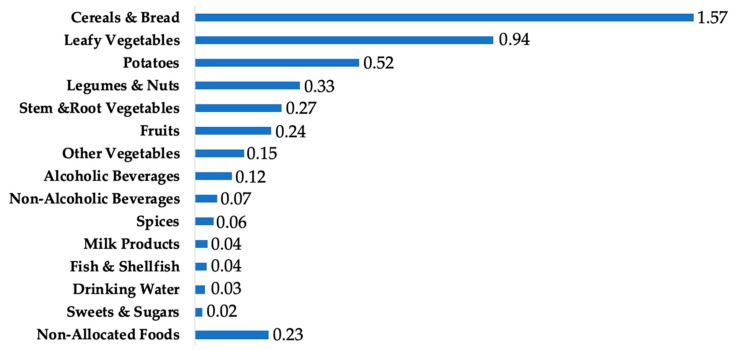
Cd contribution (µg/day/person) from major food groups among the US population aged ≥2 years in NHANES 2007–2012.

**Figure 2 nutrients-11-00002-f002:**
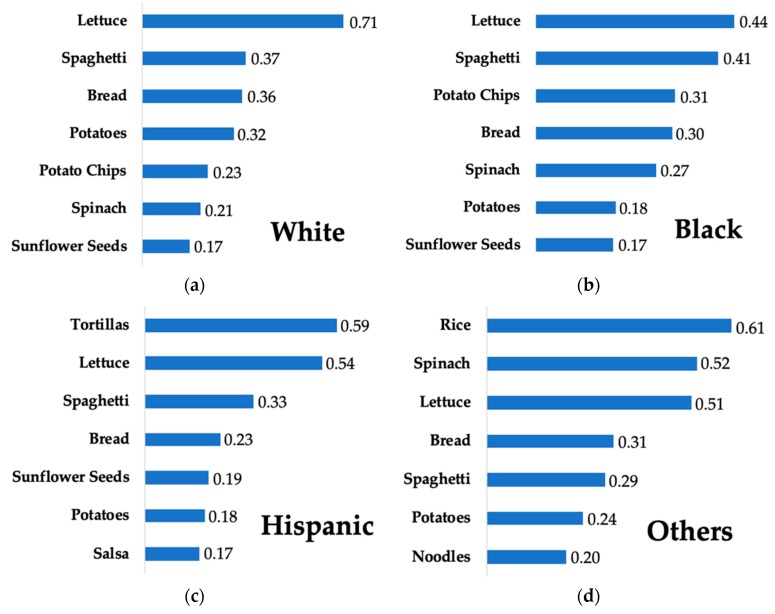
Cd contribution (µg/day/person) of top sources of dietary Cd intake among racial/ethnic groups in the US population aged ≥2 years in NHANES 2007–2012: (**a**) White; (**b**) Black; (**c**) Hispanic; (**d**) others including Asians.

**Table 1 nutrients-11-00002-t001:** Daily Cd intake and weekly Cd intake per weight among the US population aged ≥2 years by sociodemographic and lifestyle characteristics in National Health and Nutrition Examination Survey (NHANES) 2007–2012 (*n* = 12,523).

	Daily Cd Intake (µg/Day/Person)			Weekly Cd Intake Per kg Body Weight (µg/kg Body Weight/Week)		
	*n*	Mean (95% CI)	*p*-Value	*n*	Mean (95% CI)	*p*-Value
All	12,523	4.63 (4.50, 4.75)		12,411	0.54 (0.52, 0.55)	
Age (year)			<0.0001			<0.0001
2–10	3024	2.96 (2.83, 3.10)		3007	0.94 (0.89, 0.99)	
11–19	1882	4.06 (3.85, 4.27)		1857	0.49 (0.47, 0.51)	
20–30	1319	5.04 (4.68, 5.40)		1314	0.49 (0.45, 0.52)	
31–50	2521	5.34 (5.07, 5.61)		2504	0.48 (0.45, 0.51)	
51–70	2395	5.01 (4.77, 5.26)		2378	0.44 (0.42, 0.47)	
70+	1382	4.39 (4.22, 4.57)		1351	0.43 (0.41, 0.44)	
Gender			<0.0001			<0.05
Male	6463	5.09 (4.92, 5.26)		6415	0.55 (0.53, 0.57)	
Female	6060	4.15 (3.99, 4.31)		5996	0.52 (0.50, 0.54)	
Body Mass Index (kg/m^2^)			<0.0001			<0.0001
BMI < 18.5	2980	3.24 (3.04, 3.45)		2868	0.95 (0.90, 1.00)	
18.5 ≤ BMI < 25	3744	4.92 (4.69, 5.15)		3744	0.57 (0.55, 0.60)	
25 ≤ BMI < 30	2987	4.98 (4.75, 5.20)		2987	0.44 (0.42, 0.46)	
BMI ≥ 30	2812	4.78 (4.55, 5.00)		2812	0.34 (0.32, 0.35)	
Ethnicity			0.078			<0.001
White	5564	4.73 (4.58, 4.89)		5509	0.52 (0.51, 0.54)	
Black	2416	4.13 (3.91, 4.36)		2386	0.50 (0.47, 0.53)	
Hispanic	2221	4.33 (3.98, 4.68)		2207	0.55 (0.52, 0.58)	
Others	2322	4.65 (4.36, 4.95)		2309	0.64 (0.58, 0.70)	
Poverty income ratio			<0.0001			0.29
(PIR) < 1.3	4098	4.00 (3.85, 4.16)		4048	0.54 (0.52, 0.56)	
1.3 ≤ PIR < 1.85	1492	4.22 (3.90, 4.53)		1475	0.51 (0.47, 0.54)	
1.85 ≤ PIR < 3.5	2689	4.44 (4.24, 4.64)		2671	0.53 (0.50, 0.56)	
PIR ≥ 3.5	3261	5.18 (4.95, 5.42)		3243	0.55 (0.52, 0.58)	
Alcohol consumption			<0.0001			<0.0001
None	7670	3.92 (3.81, 4.02)		7584	0.60 (0.58, 0.62)	
Moderate	2494	5.50 (5.24, 5.77)		2481	0.50 (0.47, 0.52)	
Heavy	2359	5.14 (4.90, 5.39)		2346	0.47 (0.44, 0.50)	
Education level			<0.0001			0.57
Less than high school	5048	3.89 (3.75, 4.04)		5004	0.53 (0.51, 0.55)	
High school equivalent	1838	4.63 (4.40, 4.86)		1815	0.41 (0.39, 0.43)	
College	2210	4.94 (4.70, 5.18)		2192	0.45 (0.43, 0.48)	
Graduate	1863	5.95 (5.56, 6.33)		1846	0.55 (0.51, 0.59)	
Smoking status			<0.05			0.28
Current smokers	1489	4.67 (4.43, 4.91)		1477	0.43 (0.41, 0.46)	
Non-smokers	4087	5.21 (4.96, 5.45)		4049	0.48 (0.45, 0.51)	
Passive smokers	1758	4.98 (4.68, 5.27)		1738	0.44 (0.40, 0.48)	
Tap water source			0.18			0.70
Community supply	8338	4.67 (4.53, 4.80)		8265	0.54 (0.52, 0.56)	
Well or spring	1372	4.68 (4.38, 4.99)		1359	0.52 (0.49, 0.56)	
Don’t drink tap water	2326	4.43 (4.11, 4.75)		2306	0.55 (0.52, 0.59)	

**Table 2 nutrients-11-00002-t002:** Top contributing food items to total Cd intake in the US population by age group in NHANES 2007–2012.

2–10 Years Old		11–19 Years Old		20+ Years Old	
Food Item	Cd µg/Day (% Total)	Food Item	Cd µg/Day (% Total)	Food Item	Cd µg/Day (% Total)
Spaghetti	0.319 (10.8%)	Lettuce	0.477 (11.8%)	Lettuce	0.777 (15.3%)
Bread	0.267 (9.0%)	Spaghetti	0.425 (10.5%)	Spaghetti	0.357 (7.0%)
Potato chips	0.179 (6.0%)	Bread	0.307 (7.6%)	Bread	0.351 (6.9%)
Lettuce	0.163 (5.5%)	Sunflower seeds	0.274 (6.7%)	Potatoes	0.328 (6.5%)
Potatoes	0.156 (5.3%)	Potato chips	0.248 (6.1%)	Spinach	0.297 (5.9%)
Peanuts	0.145 (4.9%)	Potatoes	0.212 (5.2%)	Potato chips	0.225 (4.4%)
Noodles	0.132 (4.5%)	Tortillas	0.172 (4.2%)	Tortillas	0.170 (3.4%)
Cookies	0.131 (4.4%)	Noodles	0.171 (4.2%)	Sunflower seeds	0.163 (3.2%)
Strawberries	0.118 (4.0%)	Cookies	0.122 (3.0%)	Rice	0.159 (3.1%)
Tortillas	0.114 (3.8%)	Rice	0.117 (2.9%)	Tomatoes	0.153 (3.0%)
Milk	0.105 (3.6%)	Peanuts	0.113 (2.8%)	Noodles	0.146 (2.9%)
Rice	0.098 (3.3%)	Spinach	0.109 (2.7%)	Peanuts	0.143 (2.8%)
Spinach	0.086 (2.9%)	Strawberries	0.090 (2.2%)	Beer	0.135 (2.7%)
Apple juice	0.081 (2.7%)	Milk	0.078 (1.9%)	Onions	0.122 (2.4%)
Cereals	0.070 (2.4%)	Onions	0.070 (1.7%)	Celery	0.113 (2.2%)
Sunflower seeds	0.054 (1.8%)	Ketchup	0.068 (1.7%)	Cereals	0.100 (2.0%)
Ketchup	0.043 (1.5%)	Tomatoes	0.068 (1.7%)	Strawberries	0.099 (2.0%)
Pancakes	0.042 (1.4%)	Cereals	0.067 (1.6%)	Salsa	0.089 (1.8%)
Onions	0.039 (1.3%)	Salsa	0.060 (1.5%)	Cookies	0.088 (1.7%)
Tomatoes	0.038 (1.3%)	Celery	0.057 (1.4%)	Carrots	0.045 (0.9%)

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
