# Peer review of "Dietary Cadmium Intake and Sources in the US"

_nutrients, 2018, doi:10.3390/nu11010002_

Reviewer 1 Report

Manuscript ID: nutrients-407773

This article is an “in silico” analysis of cadmium intake and sources in the US population obtained from crossing the National Health and Nutrition Examination Survey database with the Total Diet Study database which provides Cd concentrations of a wide panel of foods.

As such, this work gives a new and interesting view of human Cd contamination in the US and is of some value to have a clearer vision of the main Cd sources of this contamination.

However, we see at least two points that deserve clarifications and comments:

1- This study reports that smokers and passive smokers have a lower daily Cd intake as compared to non-smokers. This is very surprising and in direct contradiction to many other studies, in which blood Cd was shown to go up 4 folds in smokers as compared to non-smokers. See for example:

C. Freire, R.J. Koifman, D. Fujimoto, de Oliveira, V.C. Souza, F. Barbosa Jr., S. Koifman Reference values of cadmium, arsenic and manganese in blood and factors associated with exposure levels among adult population of Rio Branco, Acre, Brazil Chemosphere, 128 (2015), pp. 70-78).

Since the authors results are unexpected, a very convincing explanation is needed.

Similarly, an explanation is needed for the surprising almost three fold increase in calculated weekly Cd intake in underweight people as compared to obese people, since underweight people are expected to eat at least proportionally less than obese people.

2- While cereals and vegetables are without any questions the highest contributors to Cd intake in the population studied, there are only two brief mentions in the article of why this is the case (the first introductory sentence, and the presence of Cd in fertilizers).  As scientists, presentation of facts is of course fundamental. But we also have a social responsibility in the possible misinterpretation of such data by the public through first degree reports citing our work. Such as, for example: “Cereals, vegetables and fresh fruits are bad for you because of their high Cd contents”… An unambiguous statement in the conclusion is needed to insist that it is of course not the food by itself that is bad, but the Cd contamination, mostly resulting from human activity. Reigning in Cd contamination is needed to improve human health, not diminishing cereals and fruits consumption...

Author Response

Thank you for reviewing our paper and for your comments. Our study examines dietary exposure to Cd. Therefore, it does not account for Cd exposure through smoking or other sources. We noted in the introduction that tobacco smoke is an important source of Cd exposure among smokers (lines 39-40), but we state that our objective is to examine dietary exposure level and sources in the US population (lines 61-64). The study by Freire et al examined blood Cd concentrations, which are reflective of Cd exposure from all sources, so it is unsurprising that they found higher blood Cd concentrations among smokers compared to non-smokers. We revised the manuscript to provide more description of the association between Cd and smoking, and cited this paper along with our group’s papers relevant to this topic (lines 37-39).

In the first columns of Table 1 we show that underweight individuals consume less dietary Cd than normal weight individuals (3.24 µg compared to 4.29 µg per day on average), but on the right side of Table 1 we show that Cd intake per kg of body weight is higher among underweight individuals than among other BMI groups.

We agree that it is important to present our findings and interpretations in a socially responsible manner. When we discuss our findings about the contribution of cereals and plant products to Cd intake, we have added a sentence to reiterate that this is related to human activity through industrial and agricultural processes (lines 182-185). We also hope that by prominently mentioning this issue in the first sentence of our paper, we will avoid confusion among readers and will clearly indicate that Cd contamination of the food supply results from human activity.

Reviewer 2 Report

Page 2, lines 45-47: The authors may list or add some information about the dietary Cd contributors and identification of population groups; Becasue staple foods contibute nearly 50% of dietary Cd intake.

How the authors estimated the Cd levels in cooked food that is moistured and fat (those may change during cooking).

The authors used 154 individual food items to analysecd contents?

The authors may consider adding the health risk assessment of dietary Cd levels

The authors listed daily/weekly Cd intake; Cd level in major 15 food groups and again top contributing foods; top sources of Cd intake among racial/ethnic groups. All these information is already available/existing. This manuscript did not add any novel new information.

Author Response

Thank you for your review, and thank you for mentioning this point. We attempted to be succinct in our introduction and clearly state our study objectives without giving more background than was necessary to explain our research premise. We do discuss staple foods in different nations (lines 186-191) and in different US racial/ethnic subgroups (lines 196-202), and how that relates to the findings of this study and other studies examining dietary Cd sources.

As noted on lines 86-87, we adjusted for moisture and fat changes in cooked foods.

It is correct that we used 154 individual foods. As noted in section 2.2 of our paper, we used Cd data for individual foods reported in the TDS and matched these items to the dietary data in NHANES.

To give readers some context about health risk, we state that average Cd consumption in the US population was 22% of the established tolerable weekly intake (TWI) of 2.5 µg/kg body weight per week (lines 20-21). We also specifically mention that this figure was higher among young children (38% of TWI) and underweight individuals (38% of TWI). We present weekly intake (µg/kg body weight per week) for the total population and for numerous subgroups in Table 1, and readers can easily compare these values to the established TWI.

Prior to this study, no information has been published on the levels or sources of dietary Cd intake in a representative sample of the US population. This question has been addressed in a cohort of postmenopausal US women, but not in a larger sample of US individuals including men, and those of younger age groups. We explain some of the gaps in knowledge on lines 56-61.

Reviewer 3 Report

This is a good and concise paper that details sources of cadmium exposure in various foods consumed in the US.  This study is useful and necessary for anyone doing cadmium research.

Minor critique

The authors point out that Cd concentrations in the food supply fluctuate over time on line 161. As such, the range of years data were collected (2006 – 2013) in the total diet study should  be included on line 220 when summarizing the study.

Author Response

Thank you for reviewing our paper. We agree that it is prudent to reiterate that our findings are reflective of 2007-2012 dietary data from NHANES (matched with corresponding years from the Total Diet Study), and that dietary exposure to Cd may change over time due to changing dietary patterns and/or changing levels of Cd in various foods. We have added this to the discussion (lines 218-220).

Round  2

Reviewer 2 Report

Do the authors performed health risk assessment  of dietary Cd levels in the population studied?

Author Response

Thank you again for reviewing our manuscript and providing your feedback. In addition to estimating absolute amounts of Cd in the diet, we calculated intake on a body weight basis (µg per kg body weight per week), which can be found in Table 1. This measurement is useful for understanding health risk, as it describes the relative burden of a given amount of cadmium ingestion in relation to body mass, and can be compared to the established tolerable weekly intake (TWI) of 2.5 µg/kg bw/wk. Therefore, we hope that the data presented in Table 1 will be easily interpreted in terms of health risk by comparing to this standard. We have revised the manuscript so that dietary Cd intake of the whole population is presented as a percentage of the TWI in the text of the results section (lines 160-161) as well as the abstract (lines 20-21). We have also added a reference (Ref #25) documenting how the TWI was established.